TOPICAL REVIEW

# Lifestyle implications of the paradox and management of oxidative stress in sperm

Giuseppe T. Patané[1,2], Ruben J. Moreira[1,3] 🆔, Annamaria Russo[2], Maria L. Pereira[4,5], Pedro F. Oliveira[3], Davide Barreca[2] and Marco G. Alves[1] 🆔

[1]*Department of Medical Sciences (iBiMED), Institute of Biomedicine, University of Aveiro, Aveiro, Portugal*

[2]*Department of Chemical, Biological, Pharmaceutical and Environmental Sciences, University of Messina, Messina, Italy*

[3]*LAQV-REQUIMTE, Department of Chemistry, University of Aveiro, Aveiro, Portugal*

[4]*Department of Medical Sciences, University of Aveiro, Aveiro, Portugal*

[5]*Department of Biology, CICECO-Institute of Materials, University of Aveiro, Aveiro, Portugal*

Handling Editors: Laura Bennet & Rebecca Simmons

The peer review history is available in the Supporting Information section of this article (https://doi.org/10.1113/JP289694#support-information-section).

**Giuseppe Tancredi Patanè** is a PhD student in Biochemistry at the University of Messina. He graduated with honors in Chemistry and Pharmaceutical Technology and focuses on the biological activity of poly-phenols and their applications in human health. **Marco G. Alves** is an Assistant Professor and Principal Investigator at the University of Aveiro, with a background in Biology and a PhD in Biochemistry. His research focuses on applying omics technologies to understand various diseases, particularly male infertility. He has published over 230 scientific papers, holds two patents, and has received more than a dozen research awards. He serves on the editorial boards of over 15 international journals and has been invited to speak at more than 60 conferences worldwide. Alves has been consistently listed among the top 2% most-cited scientists globally, according to Stanford University. His recent work explores the impact of paternal lifestyle—especially nutrition—on reproductive health and offspring development.

**Abstract figure legend Lifestyle factors influencing male fertility**. Schematic overview of lifestyle factors with beneficial (left) and detrimental (right) effects on male fertility. A balanced diet, regular moderate physical activity and stress management strategies support sperm function and redox homeostasis. In contrast, exposure to environmental toxins (heavy metals, bisphenols, phthalates, microplastics), obesity, sedentary behaviour, and high-calorie diets/obesity promote oxidative stress and impair spermatogenesis.

**Abstract**   Oxidative stress (OS), defined by an imbalance between reactive oxygen species (ROS) production and antioxidant defences, plays a bivalent and paradoxical role in the male reproductive system. At physiological levels, ROS are indispensable for sperm capacitation, hyper-activation and acrosome reaction, which are crucial for fertilization. However, excessive ROS – stemming from both endogenous sources (e.g. mitochondrial metabolism, enzyme-mediated reactions, seminal leukocytes) and exogenous factors (e.g. environmental pollutants and life-style behaviours) – can trigger detrimental OS reactions, including lipid peroxidation, DNA damage and impaired sperm function, contributing to male infertility. This dualistic nature of OS complicates defining the distinction between its physiological and pathological concentrations of ROS. This review comprehensively examines the complex interplay between ROS and OS in male reproduction, delineating how lifestyle factors can contribute to this imbalance and what mechanisms are implicated. Furthermore, we discuss current and emerging non-pharmacological strategies aimed at mitigating pathological OS, including antioxidant supplementation (e.g. resveratrol, vitamins C and E, coenzyme Q10), dietary interventions such as adherence to the Mediterranean diet and lifestyle modifications like regular moderate exercise and stress management techniques. By elucidating these multifaceted aspects, our analysis provides critical insights into maintaining redox homeostasis and advancing clinical interventions for improved male reproductive health.

(Received 7 August 2025; accepted after revision 19 November 2025; first published online 16 December 2025)

**Corresponding authors** D. Barreca: Department of Chemical, Biological, Pharmaceutical and Environmental Sciences, University of Messina, Messina, Italy.     Email: davide.barreca@unime.it; Marco G. Alves: Department of Medical Sciences (iBiMED), Institute of Biomedicine, University of Aveiro, Aveiro, Portugal.     Email: marcoalves@ua.pt

## Introduction

In recent years, especially in industrialized countries, people have been exposed to an increasingly frenetic lifestyle and environmental factors that contribute to stress in every form. The so-called oxidative stress (OS) has been implicated in most diseases (Forman & Zhang, 2021). This term, introduced by Sies and co-workers in 1985, describes a condition caused by excessive production of reactive oxygen species (ROS) (Sies & Cadenas, 1985). Research in this field has grown exponentially, with over 340,000 publications on PubMed as of 2025 addressing OS and its role in health and disease. *Free radicals* are short-lived molecules ($10^{-10}$–$10^{-3}$ s) containing one or more unpaired electrons. ROS such as superoxide radicals ($O_2^{-}$), hydrogen peroxide ($H_2O_2$), hydroxyl radicals ($OH^{-}$), hypochlorous acid (HOCl) and singlet oxygen ($^1O_2$) are byproducts of cellular redox reactions. When their levels exceed antioxidant defences, OS plays an orchestral role in the etiopathogenesis of various diseases, including infertility. Male infertility affects 20%–30% of couples, and OS is expected to contribute to between 30% and 80% of cases (Magalhaes et al.,

2021). Two major sources of ROS in semen are sperm and neutrophils (Aitken, Jones et al., 2012). The latter are among the most abundant leukocytes in semen and produce high concentrations of ROS to defend the semen from pathogens. Consequently, if there is an elevated concentration of neutrophils, it leads to higher production of ROS and potentially ROS-related infertility. As early as 1943, McLeod reported that incubation of sperm in the presence of high oxygen reduced sperm motility, whereas the addition of antioxidant enzymes like catalase (CAT) counteracted the production of hydrogen peroxide, improving sperm motility (MacLeod, 1943). Particularly, the production of ROS is inversely proportional to the state of maturation of the sperm during spermatogenesis (Said et al., 2005). The sperm, once ejaculated, does not have the capacity to fertilize (sperm–oocyte fusion) until it completes the phenomenon of 'capacitation'. This term indicates a series of biochemical events such as a decrease in membrane cholesterol, removal of inhibiting factors, increase in permeability, increased $Ca^{2+}$ and $HCO_3^{-}$, relative increase in pH, activation of second messengers such as soluble adenylyl cyclase (sAC), production of cAMP and consequent phosphorylation

of tyrosine residues. This cascade of events causes an increase in sperm motility, known as 'hyperactivation', the binding of sperm to eggs and 'the 'acrosome reaction' as a result of increased sperm motility (Carrageta et al., 2020). Early studies by Jones et al. and de Lamirande and Gagnon revealed that due to their high polyunsaturated fatty acid content, sperm membranes easily undergo lipid peroxidation, reducing membrane fluidity and motility (de Lamirande & Gagnon, 1992; Jones et al., 1979). ROS affects not only the plasma membrane but also the sperm DNA, causing multiple fragmentations and decreased sperm quality (Aitken, 1999). Overall unbalanced OS compromises motility and vitality, resulting in lower fertility capacity. Although OS is potentially linked to male infertility, it also plays an essential role in sperm physiology. This work aims to discuss the dual role of OS in sperm cells, with particular emphasis on endogenous and exogenous factors that change ROS balance in the reproductive system and sperm quality.

## Endogenous sources of ROS that affect sperm physiology

As previously mentioned, the first evidence of the presence of ROS and OS in human sperm dates to 1943 (MacLeod, 1943). About 30 years later the first correlation between ROS-induced lipid peroxidation and potential male infertility was reported (Jones et al., 1978). Since then, numerous studies have investigated endogenous and exogenous ROS production in human sperm and the male reproductive tract, as shown in Fig. 1. Spermatozoa are specialized cells that produce high levels of ATP to perform their functions. From a biochemical point of view, their energy production is mainly carried out via glycolysis and oxidative phosphorylation (OXPHOS), although they present great metabolic plasticity depending on the situation. During OXPHOS, as protons are pumped into the inner mitochondrial membrane to drive ATP synthase to produce ATP, ROS are produced primarily at complexes I and III of the

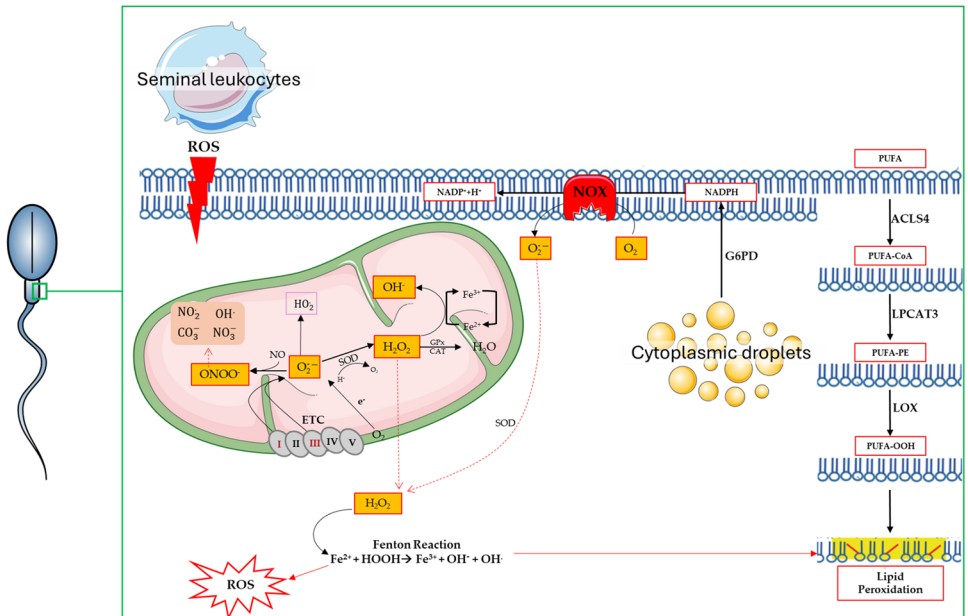

**Figure 1. Schematic representation of the main sources of ROS (reactive oxygen species) in human spermatozoa**

Leukocytes, present in seminal fluid, produce high concentrations of ROS, which trigger lipid peroxidation, irreversibly damaging membrane permeability and integrity. In this context the enzyme acyl-coenzyme A synthetase long-chain family member 4 (ACSL4) catalyses the addition of coenzyme A (CoA) to PUFAs (polyunsaturated fatty acids) to produce PUFA-CoA, which becomes a substrate for lysophosphatidylcholine acyltransferase 3 (LPCAT3) to produce PUFA-PE, which are then subjected to the action of lipoxygenase (LOX), which adds oxygen in a stereospecific manner, to produce PUFA-OOH. In spermatozoa, the cytoplasmatic droplets contain glucose-6-phosphate dehydrogenase (G6PD), which is involved in glycolysis, in the intracellular production of the reduced form of NADPH, which in turn is a substrate of NADPH oxidase (NOX), contributing to the production of superoxide radicals ($O_2^-$). Internal sources of ROS also include the mitochondria, in particular the electron transport chain (ETC), where complexes I and III produce superoxide ($O_2^-$) as a byproduct of oxidative metabolism. $O_2^-$ can react directly with nitric oxide (NO) to form peroxynitrite ($ONOO^-$) but can be rapidly converted to hydrogen peroxide ($H_2O_2$) through the action of superoxide dismutase (SOD), which, if not neutralized by CAT (catalase)/GPX (glutathione peroxidase), can contribute to the production of hydroxyl radical (OH–) via the Fenton and Haber–Weiss reaction in presence of ion ($Fe^{2+}$).

electron transport chain (ETC). This delicate system balances ATP production with ROS production and is sensitive to a variety of factors, including temperature, understood as a source of energy that can stimulate mitochondrial activity but can also lead to excessive heat and consequently to mitochondrial dysfunction. Furthermore, after the production of ROS, highly electrophilic aldehyde byproducts such as malondialdehyde (MDA), acrolein, and 4-hydroxynonenal (4-HNE) are formed. These compounds can bind to mitochondrial proteins, including succinic acid dehydrogenase, blocking ATP formation and exacerbating OS and lipid peroxidation (Aitken, Jones et al., 2012). Correlation between sperm morphology and ROS production was first described in 1996, with morphologically normal spermatozoa exhibiting lower production of free radicals. More recently, the term 'sperm deformity index' (SDI) was introduced, which quantitatively represents the number of sperm deformities and thus indirectly predicts ROS production in sperm with borderline morphology (e.g., acrosomal damage, cytoplasmatic droplets, amorphous heads) (Walczak-Jedrzejowska et al., 2013). During spermiogenesis, sperm elongation and a change in the plasma membrane occur, with a lower presence of docosahexanoic acid (DHA). DHA represents 43% of the total membranes' polyunsaturated fatty acid (PUFA) concentration, leading to decreased lipid peroxidation and lower cytoplasmatic content. Incomplete maturation results in spermatozoa with abnormal morphology, increased cytoplasmic retention, and the accumulation of the enzyme glucose-6-phosphate dehydrogenase (G6PD). This enzyme produces 6-phosphoglucunate and NADPH, which is an essential substrate for NADPH oxidase (NOX) that produces ROS. Consequently, men with teratozoospermia exhibit a higher release of immature spermatozoa characterized by biochemical markers of cytoplasmic retention, like G6PD, creatine kinase (CK), and lactate dehydrogenase (LDH), resulting in lower fertilizing potential (Said et al., 2005). Leukocytes present in seminal fluid play a crucial role in protecting against bacterial infection and inflammation, but they produce up to 1000 times more ROS than spermatozoa. This substantial production of ROS, which intends to protect nearby organs, negatively impacts sperm quality. Human male samples with bacterial infections have a higher percentage of spermatozoa that initiate apoptosis and biochemical markers of cell death, including increased mitochondrial membrane potential ($\Delta\Psi m$), caspase-3/7 activation, and DNA fragmentation in ejaculated sperm samples (Mupfiga et al., 2013).

## How lifestyle impacts exogenous sources of ROS that affect sperm physiology

Although the impact of endogenous sources of OS is well known, a growing body of evidence highlights the significant contribution of numerous environmental and exogenous factors related to modern lifestyle. These factors can have a significant impact on semen quality. The major risk factors include alcohol and drug use, cigarette smoking, exposure to heavy metals and environmental pollutants, certain medical conditions and dietary habits, as shown in Fig. 2.

**Diet.** Cristodoro and co-workers, in a 2024 work, demonstrated the influence of diet on sperm quality by demonstrating how specific dietary patterns can positively or negatively modulate semen quality (Cristodoro et al., 2024). There is a close correlation between sperm cell health and food intake, particularly the necessary balance between PUFAs and saturated fatty acids (SFAs). The PUFAs that we mostly obtain from food are linoleic acid and $\alpha$-linolenic acid, which through a series of enzymatic reactions catalysed by $\Delta6/5$-desaturase result in n-6 and n-3 PUFA derivatives such as DHA, eicosapentaenoic acid (EPA), arachidonic acid (ARA), and dimo-$\gamma$-linolenic acid (DGLA). Although the optimal substrate for $\Delta6$-desaturase is $\alpha$-linolenic acid, unhealthy Western diets usually contain more linoleic acid, which leads to a higher production of omega-6 and a higher production of ARA, potentially impacting the fatty acid profile of sperm cells (Calder, 2017). ARA, released from plasma membranes, can be converted by enzymes like lipoxygenases (LOXs) and cyclooxygenases (COXs) into eicosanoids, molecules that can promote inflammation. In fact subjects on a Western diet have an average of 10 times more linoleic acid intake than $\alpha$-linolenic acid (Saez & Drevet, 2019). A diet rich in fat and calories results in SFA, accounting for ~60% of the total fat intake. This high level of SFA intake correlates with an elevated ARA concentration, plasma membrane disruption, reduced capacitation, increased OS, and lipid peroxidation. In Western countries where there is a higher incidence of dyslipidaemia and hypercholesterolaemia, there is also an increase in oligo/asthenozoospermia, that is, patients with normal sexual function and ejaculation but with a sperm concentration <20 million/mL and reduced motility, respectively (Comhaire & Mahmoud, 2006). Therefore, these findings suggest an association between high SFA intake and impaired sperm parameters. However due to the observational nature of the available

studies and the potential presence of confounding factors, a precise correlation cannot be definitively established. In addition, those diets are associated with reduced DHA and increased omega-6-to-omega-3 fatty acid ratio, production of ROS, and lipid peroxidation (Salvaleda-Mateu et al., 2024). Furthermore, a diet rich in SFAs was reported to negatively impact the energy balance of sperm cells. After only 4 weeks on a high-fat diet (35.2% fat, 36.1% carbohydrate), mice exhibited an increase in weight and a 30% inhibition in seminal LDH-C4 activity. As a consequence, there is a lower production of $NAD^+$ and consequently reduced glycolysis and Krebs cycle activity (Ferramosca et al., 2016). In addition, a diet lacking fruits and vegetables has detrimental effects on male fertility by limiting the intake of vitamins and natural antioxidants that are essential for the balance of ROS. For example, supplementation with antioxidants (e.g., vitamins C and E) can interrupt the radical chain of membrane phospholipid oxidation before it damages the mitochondria, ultimately improving sperm motility and quality (Suleiman et al., 1996). Consuming fresh fruits and vegetables also provides a source of daily intake of zinc that plays a key role in male fertility by facilitating folic acid absorption, acting as an antioxidant, and exhibiting anti-apoptotic properties. It inhibits caspases in early apoptotic pathways and blocks magnesium- and calcium-dependent endonucleases, preventing DNA fragmentation (Chimienti et al., 2003). This was further supported by a recent meta-analysis showing that zinc

and folate supplementation improves sperm quality (e.g., morphology, count, and motility) (Irani et al., 2017). Conversely, the consumption of foods and drinks rich in simple sugars is correlated with an increase in glycaemic index, spermatozoa inflammation, and dysregulated glucose metabolism (Adekunbi DA et al., 2016). In individuals with diabetes, elevated advanced glycation end (AGE) binding to their receptors (RAGE) in the testes and epididymis activates nuclear factor kappa B (NFkB), leading to increased NADPH and ROS production, exacerbating inflammation and DNA fragmentation in spermatozoa (Karimi et al., 2011). Several studies using *in vitro* glycation models, where spermatozoa were incubated with glycating compounds (mannitol, glucose), demonstrated the increased production of carboxymethyl-lysine (CML) and 8-oxoguanine, markers of AGE-induced damage to mitochondrial and sperm DNA (Nevin et al., 2018; Wang et al., 2009). Hyperglycaemia and dyslipidaemia are strongly associated with obesity and elevated body mass index (BMI). In men with overweightor obesity, increased white adipocyte tissue leads to overproduction of pro-inflammatory cytokines, including interleukins (IL) IL-1, IL-6 and IL-18, and tumour necrosis factor alpha (TNF-$\alpha$). The latter also exhibits metabolic effects by reducing testosterone production through affecting the luteinizing hormone (LH) (Sullivan, 2015). This pro-inflammatory state causes the destruction of the epididymal epithelium, further recruiting macrophages and neutrophils, which

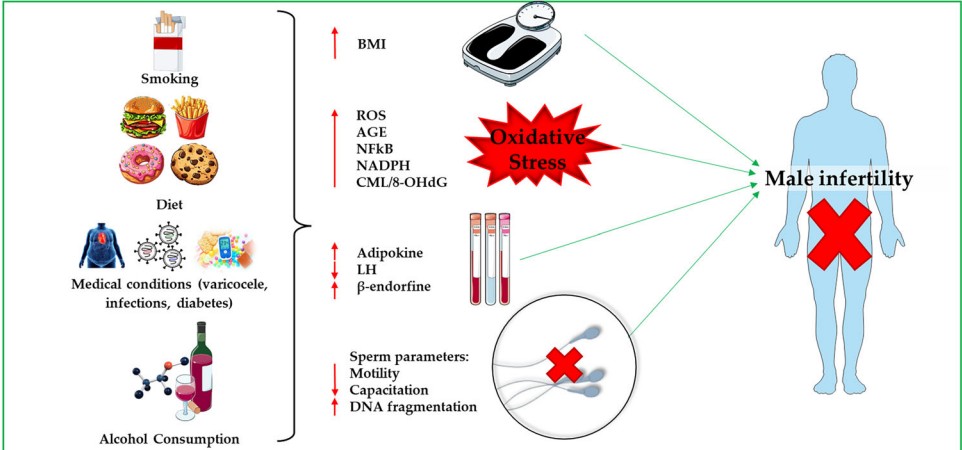

**Figure 2. Schematic representation of factors contributing to male infertility through the induction of OS (oxidative stress) and hormonal imbalances**

High tobacco consumption; an unbalanced diet rich in processed sugars, fried foods and excess meat; medical conditions such as obesity (with the associated increase in body mass index (BMI)), varicocele and infections; and alcohol consumption are risk factors for male infertility. Clinically, OS is characterized by elevated production of reactive oxygen species (ROS), accumulation of advanced glycation end (AGE) products, activation of the transcription factor NFkB (nuclear factor kappa B), and increased NADPH, which consequently increase NADPH oxidase (NOX) activity. Hormonal changes are also recorded, including an increase in adipokines such as leptin and $\beta$-endorphins and a decrease in luteinizing hormone (LH). These processes lead to reduced motility and capacitation and increased DNA fragmentation, impairing sperm quality and functionality and contributing overall to male infertility.

contributes to an increase in ROS and OS, thus resulting in reduced sperm motility, impaired acrosomal reaction, and increased DNA damage (Bisht et al., 2017). Testosterone and LH production decrease due to increased aromatase activity, which converts testosterone into oestrogen. In addition, inflammatory and oxidative processes are fuelled by adipokines, such as leptin, resistin, chemerin, ghrelin, and adiponectin. Leptin, also known as satiety hormone, seems to have a contradictory effect: although it can improve sperm capacitation and acrosine activity through activation of the STAT3-ERK1/2 pathway, its overproduction in subjects with obesity has been hypothesized to reduce acetate production by Sertoli cells, leading to decreased sperm motility and increased infertility (Martins et al., 2015). Conversely, adiponectin, mainly present in its high-molecular-weight (HMW) form in germ cells and Leydig cells, is negatively correlated with increased BMI and appears to play a positive role in determining sperm quality and quantity (Elfassy et al., 2018). From a molecular point of view, Mayank and co-workers recently demonstrated that adiponectin stimulates spermatogenesis by inhibiting caspase-3, increasing the expression of proliferating cell nuclear antigen (PCNA) and B-cell lymphoma 2 (Bcl2), enhancing AKT phosphorylation and subsequently upregulating the expression of GLUT8, which leads to better uptake of glucose (Choubey et al., 2019). Finally scrotal fat and adipocyte accumulation increase scrotal temperature, further impairing the temperature-sensitive process of spermatogenesis.

**Smoking.** Cigarette smoking is recognized as a major factor affecting semen quality, though in the past studies yielded conflicting results due to methodological variations and confounding variables (Mostafa, 2010; Sepaniak et al., 2006). Smoking metabolites, such as benzo[a]pyrene-diol-epoxide-DNA adducts, are found in the spermatozoa of smokers but not in healthy men (Ranganathan et al., 2019). Nicotine, among the 4000 compounds that are released by smoking, disrupts the sperm membrane, leading to DNA double-strand breaks and increased ROS production (Arabi, 2004). Nicotine metabolites in the seminal fluid of smokers are correlated with elevated levels of the mutagen 8-hydroxy deoxyguanosine (8-OHdG), which induced high caspase-3 activation and DNA damage (Ranganathan et al., 2019). To prevent the oxidants contained in cigarette smoke and to prevent the formation of ROS and OS, smokers may require a 2- to 3-fold increase in dietary intake of natural antioxidants (Fraga et al., 1996). Furthermore, approximately 60% of non-smokers are exposed to second-hand smoke, comprising mainstream smoke (MS), exhaled by the smoker, and sidestream smoke (SS), released from the burning cigarette (Shields, 2007). Research in mouse models has demonstrated that SS significantly contributes to sperm damage, including DNA fragmentation, reduced fertilization capacity, chromatin damage, and decreased motility (Marchetti et al., 2011). SS exposure may also have transgenerational effects, such as mutations in expanded simple tandem repeats (ESTRs), suggesting that passive smoking acts as a human germ cell mutagen (Omolaoye et al., 2022; Pereira et al., 2014). In sum the detrimental effect of smoke extends beyond basic semen parameters to include severe DNA damage, likely mediated by smoking-related metabolites and increased OS. Furthermore, the dangers of second-hand smoke, especially SS, cannot be ignored, as it also negatively impacts sperm quality and may even induce heritable genetic changes. These findings underscore the importance of policies promoting smoking cessation and the avoidance of passive smoke exposure for men's reproductive well-being.

**Alcohol consumption.** Human and animal studies have shown that alcohol affects spermatogenesis (Organization, 2024). It significantly affects spermatogenesis and semen quality through various mechanisms, including azoospermia, endocrine disruption through alterations on the hypothalamic–pituitary–gonadal (HPG) axis and testicular function (van Thiel et al., 1979). Within the testes, alcohol consumption correlates positively with $\beta$-endorphin release, which in turn lowers the expression of luteinizing hormone-releasing hormone (LHRH) receptors, reduces testosterone production and causes apoptosis of Leydig and seminiferous cells, leading to an overall impairment of spermatogenesis (Gianoulakis, 1990; Maneesh et al., 2006). Furthermore, alcohol consumption induces OS in testicular cells, characterized by decreased $\Delta\Psi m$, glutathione (GSH) levels and the activity of glutathione peroxidase (GPx), but an increase in caspase-3, p53, Bax-to-Bcl2 ratio, translocation of cytochrome c, ROS production and subsequent cell death. However, evidence regarding the impact of alcohol consumption on sperm DNA fragmentation remains inconclusive (Condorelli et al., 2015; Nguyen-Thanh et al., 2023). Although the literature consistently reports reduced semen volume and morphological abnormalities in habitual alcohol users, the effects on sperm motility and density in occasional drinkers are less clear (Ricci et al., 2017; Ricci et al., 2018). Overall, alcohol consumption remains a threat to male reproductive health by affecting spermatogenesis, hormone balance and even the function of testicular cells. The detrimental effects of chronic alcohol consumption on sperm parameters (e.g., volume, morphology) are well established, although whether OS is the underlying link remains a matter of debate. The impact of alcohol on sperm DNA integrity, a phenomenon intricately linked to OS, also remains elusive. Finally, how occasional alcohol consumption affects male reproductive health deserves special attention.

## Antioxidant defence mechanisms protecting sperm in brief

The previous discussion highlighted both endogenous and exogenous sources of ROS and OS in spermatozoa. During maturation and cytoplasm shrinkage, spermatozoa lose much of their cytoplasm, resulting in decreased ability to combat oxidative damage. Key antioxidant systems present in spermatozoa, although in limited quantities, include superoxide dismutase 1/2 (SOD1/2), CAT, various enzymatic isoforms of GPx, thioredoxin (TRX) and peroxiredoxin (PRDX). Of these, GPx4 plays a crucial role during maturation, particularly in the mitochondria, where it contributes to a sheath essential for mitochondrial construction; its dysfunction has been associated with male infertility (Fujii et al., 2003). PRDX6, through its lysophosphatidylcholine acyltransferase activity, can replace these damaged phospholipids, thus playing a vital role in membrane repair. The combined activity of SOD, CAT and GPx, produced in the testes and prostate, is crucial for managing the leukocyte-derived ROS. Interestingly germ cells, under certain conditions, can stimulate SCs through IL-1$\alpha$ to release SOD-3 into the seminal fluid to increase antioxidant defences (Aitken & Roman, 2008). Particularly $O_2^{.-}$, produced primarily by SOD-1 activity and to a lesser extent extracellular SOD-3, is converted to hydrogen peroxide ($H_2O_2$). Subsequently, GPx and CAT work together to detoxify $H_2O_2$, converting it into water and molecular oxygen, thus completing the ROS detoxification pathway. In conclusion, although spermatozoa possess a limited level of antioxidant enzymes, these systems, along with contributions from the surrounding seminal fluid, are crucial for maintaining sperm health. The interplay of SOD, CAT and GPx is essential for controlling ROS levels and preventing oxidative damage, highlighting the delicate balance required for proper sperm function and male fertility.

## Non-enzymatic antioxidants

In the spermatozoa, production of ROS can also be balanced and neutralized by the presence of non-enzymatic antioxidant systems. For example $\alpha$-tocopherol (vitamin E), a lipophilic compound particularly abundant in fruit- and vegetable-rich diets, integrates into the phospholipid bilayer of sperm membranes, halting the ROS-induced lipid peroxidation chain. *In vivo* studies have shown that vitamin E supplementation can improve sperm motility and decrease MDA levels, which are directly linked to OS in membranes (Gvozdjakova et al., 2015). Furthermore, studies in mice demonstrated that combined supplementation with vitamin E and Se (another natural antioxidant present in spermatozoa) can improve semen parameters and restore normal testicular and epididymis weight when mice were exposed to several concentrations of deltamethrin, a synthetic insecticide that produces ROS and has a negative impact on male fertility (Oda & El-Maddawy, 2012). However, the doses of deltamethrin used in this *in vivo* study were significantly higher than typical human exposure levels, which limits the direct translation of these findings to humans. Ascorbic acid (vitamin C) due to its hydrophilic nature effectively balances the production of ROS, OS and the DNA fragmentation that they generate. GSH is one of the most active non-enzymatic antioxidants within the spermatozoa. It is a tripeptide formed from cys, glu and gly through the action of the enzymes glutathione synthetase (GS) and $\gamma$-glutamylcysteinesynthetase ($\gamma$GCS). Its sulphydryl group protects cells from OS. In presence of ROS and free radicals, two molecules of GSH are oxidized to one molecule of oxidized glutathione GSSG, which is subsequently transformed back into the reduced form by glutathione reductase (GR) and NADPH. Thus, the ratio of GSH/GSSG serves as an indicator of the oxidative balance within spermatozoa. Consistent with this a study using semen from 92 donors (fertile/unfertile) showed a negative correlation between infertility and GSH content in spermatozoa, with fertile men exhibiting significantly higher GSH levels compared to oligo/asthenozoospermia subjects (Fafula et al., 2017). Coenzyme Q10 (CoQ10), or ubiquinone, is another potent non-enzymatic antioxidant that is highly concentrated in sperm cells, particularly within mitochondria, and works to decrease total ROS levels and improve semen motility and concentration in men with idiopathic oligo/asthenoteratospermia (Alahmar et al., 2021). Carotenoids are also present in seminal fluid and play a key role in balancing OS. Of these lycopene, which is distributed and abundant in the testis and prostate, possesses a powerful scavenger action reportedly 10 times greater than vitamin E and even greater than $\beta$-carotene (Vakili et al., 2024). Clinical studies have further demonstrated that blood carotenoid levels may serve as potential biochemical biomarkers for predicting sperm quality, OS and ROS levels in sperm cells (Benedetti et al., 2012). However, this approach remains largely experimental and is not routinely applied in clinical settings. In conclusion, the combined actions of enzymatic and non-enzymatic antioxidant systems are essential for maintaining sperm health and function. These systems work synergistically to control ROS levels and minimize damage due to OS, highlighting the importance of a balanced redox environment for optimal male fertility. Dietary intake of vitamins, minerals and other antioxidants plays a crucial role in supporting these vital defence mechanisms.

## The dual role of OS in sperm

**Physiological roles of ROS.** MacLeod and Jones were among the first to point out that sperm cells produce ROS and free radicals and that they can have a detrimental effect on sperm quality, impacting motility and membrane stability. However emerging works emphasized how low levels of ROS appear to be essential for spermatozoa capacitation, hyperactivation, acrosome reaction and fusion of spermatozoa to oocyte. These physiological levels of ROS, primarily produced by mitochondria and sperm-specific oxidases, are essential for several molecular events, including membrane fluidity, opening of calcium channels, increased pH and tyrosine phosphorylation, confirming how they play a dual role in the spermatozoon (Tremellen, 2008). Mitochondria and oxidases from spermatozoa produce $O_2^{\cdot-}$ which react with –NO forming ONOO–, which converts membrane cholesterol to oxysterols, facilitating their release from the membrane and increasing their fluidity. During capacitation ROS are involved in the redox regulation of tyrosine phosphatase activity and adenylate cyclase (AC) activation, leading to cAMP production and activation of protein kinase A(PKA), resulting in positive modulation of specific kinases (pp60cSRC/cABL) that causes the phosphorylated state of tyrosines. This process is essential for the binding between sperm membrane and zona pellucida ZP-3 protein (Aitken, 2017). In addition to the AC/PKA pathway, ROS influence other signalling cascades, including the nitric oxide (NO)-dependent stimulation of RAS/ERK pathways, and cause the phosphorylation of Thr-Glu-Tyr motif in the spermatozoa tail and ultimately tyrosine kinase (PTK) activation (O'Flaherty et al., 2006). The important and positive role of ROS in capacitation is further supported by *in vitro* studies showing that antioxidant enzymes like SOD and CAT prevented capacitation by blocking the positive role of ROS (O'Flaherty et al., 2003). ROS are also implicated in the stimulation of sperm cell motility (hyperactivation), because the production of $O_2^{\cdot-}$ via xanthine/xanthine oxidase promotes hyperactivation, whereas antioxidant supplementation (e.g. SOD and CAT) drastically reduces sperm motility (de Lamirande & Gagnon, 1993). During the acrosome reaction ROS levels significantly increase, highlighting their importance for fertilization (de Lamirande et al., 1998). Studies using ROS inducers (e.g. A23187, NADPH or fetal cord blood serum ultrafiltrate (FCSu)) and antioxidants have further dissected the specific roles of different ROS such as $H_2O_2$ on tyrosine phosphorylation during the acrosome reaction (Walczak-Jedrzejowska et al., 2013). In conclusion ROS, once viewed primarily as detrimental to sperm health, are now recognized as essential signalling molecules for successful fertilization. Their precise spatiotemporal regulation is crucial for orchestrating the complex molecular events underlying capacitation, hyperactivation and the acrosome reaction.

**Pathological roles of excessive ROS.** An imbalance favouring ROS production over antioxidant defences leads to OS, which is detrimental to sperm quality and fertility. This delicate balance between ROS and antioxidant defences in spermatozoa is essential because mature spermatozoa have very limited antioxidant capacity, and there is a high presence of membrane phospholipids. When this equilibrium is disturbed, endogenous and exogenous ROS attack membrane phospholipids, initiating lipid peroxidation, with the formation of peroxyl radicals (LOO°) and other reactive aldehydes like MDA, acrolein and 4-HNE, which compromise membrane structure, permeability and protein function (Aitken, Whiting et al., 2012). These lipid peroxidation subproducts can further exacerbate OS by covalently binding to mitochondrial proteins, disrupting the ETC and triggering additional ROS production. Beyond membrane damage, ROS also target DNA, causing base oxidation, fragmentation products (single-/double-strand breaks) and other lesions. These DNA lesions can be quantified by measuring the marker 8-oxoguanine and by transferase dUTP nick end labelling (TUNEL), Comet assay, sperm chromatin structure (SCS), and sperm chromatin dispersion (SCD) assays (Muratori et al., 2006). Elevated ROS production and DNA damage are often correlated with impaired sperm maturation, increased G6PD activity and a cascade of free radical production (Wang et al., 2003). Immature spermatozoa, already predisposed to apoptosis, exhibit heightened levels of apoptotic markers (e.g. Fas, cytochrome c, caspases 1, 3, 8 and 9), making them even more susceptible to OS-induced damage (Li et al., 2024). The cumulative effect of OS on sperm membranes and on DNA negatively impacts standard semen parameters (e.g. ejaculate volume, concentration, motility, morphology). Studies consistently demonstrate a correlation between lipid peroxidation and DNA fragmentation with reduced motility, concentration and morphological abnormalities, particularly in subfertile men (Saleh et al., 2003). Crucially, excessive OS compromises essential processes for fertilization: capacitation, hyperactivation and acrosome reaction. Lipid peroxidation alters membrane fluidity and integrity, hindering appropriate sperm–oocyte interaction. Membrane damage also disrupts the proper functioning of ion channels ($Ca^{2+}/Mg^{2+}$ ATPase) and ATP production and decreases axonemal protein phosphorylation, ultimately impairing spermatozoa motility and hyperactivation (de Lamirande & Gagnon, 1995). Furthermore OS can directly affect sperm–oocyte fusion by oxidizing crucial sulfhydryl groups (SH) residues on sperm surface proteins involved in this interaction. Indeed, Mammoto and co-workers

incubated spermatozoa with fixed concentrations of hypoxanthine (HX) and varying concentrations of xanthine oxidase (XO) to produce increasing concentrations of ROS and showed that high OS causes oxidation of the SH residues of some sperm proteins, which are involved in the interaction and fusion with oocytes (Mammoto et al., 1996). In conclusion, whereas ROS plays a vital signalling role in sperm physiology at low concentrations, OS, resulting from an imbalance between ROS production and antioxidant capacity, severely compromises sperm quality and male fertility. By damaging sperm membranes and DNA, OS interferes with key processes such as capacitation, hyperactivation and acrosome reaction, ultimately hindering successful fertilization (Fig. 3). In humans, elevated OS levels are observed in various clinical and environmental conditions, including varicocele, genital tract infections, inflammation, obesity, smoking

and exposure to environmental pollutants or heat stress. These situations can significantly increase ROS production beyond the antioxidant capacity of seminal plasma, thereby mirroring the high OS levels reported in experimental studies. Therefore, maintaining redox homeostasis is essential for optimal sperm function and male reproductive health.

## How lifestyle impacts the paradox of OS in sperm physiology and male fertility

**Environmental toxins.** Modern life exposes individuals daily to various environmental and lifestyle-related pollutants and toxins that are present everywhere, including in the air we breathe, in the objects we use and in the food we consume. Many of those

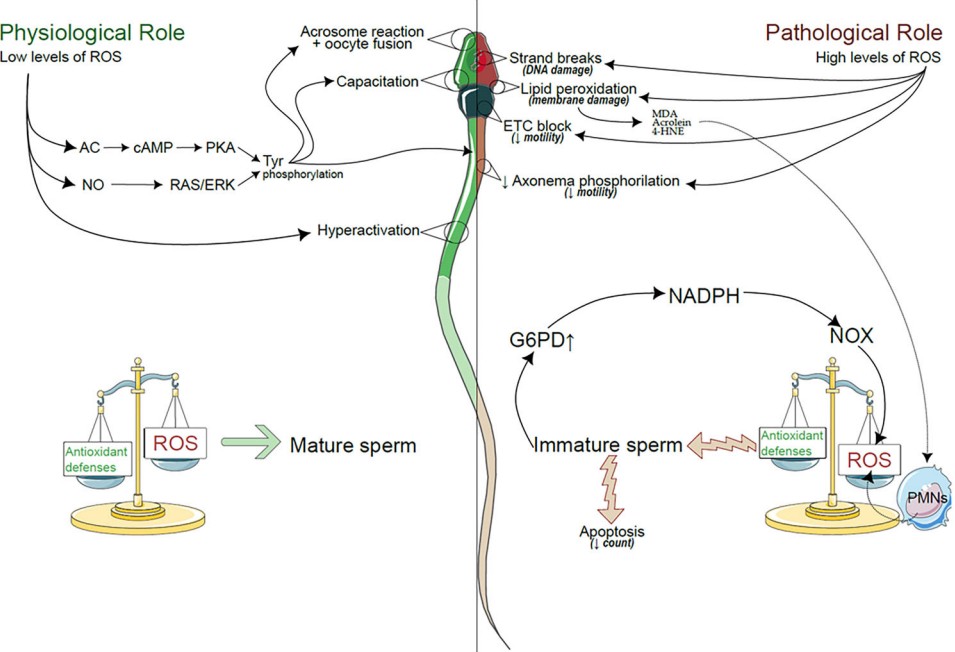

**Figure 3. The dual role of ROS in sperm function and pathology**
This figure shows the physiological and pathological roles of ROS in sperm function. The left side (green, 'physiological role') illustrates the beneficial effects of low ROS levels, whereas the right side (red, 'pathological role') depicts the detrimental effects of excessive ROS concentration. Physiological role (left panel): low levels of ROS allow essential sperm functions, including capacitation, hyperactivation and acrosome reaction, which are pivotal for fertilization. Low levels of ROS facilitate the activation of AC that increases cAMP, leading to positive modulation of PKA and tyrosine phosphorylation, which promotes sperm–oocyte fusion. Furthermore, the NO-dependent pathways (RAS/ERK signalling) support sperm motility, particularly the phosphorylation of Thr-Glu-Tyr motif in the tail of spermatozoa. A balance between ROS and antioxidant defence (SOD, CAT) helps regulate ROS levels to ensure sperm maturation and proper function. Pathological role (right panel): elevated ROS levels result in OS, leading to lipid peroxidation (increasing membrane damage), DNA fragmentation and impaired sperm motility by blocking ECT and consequently reducing tyrosine phosphorylation. The presence of immature sperm, with increased retention of G6PD, produces excess NADPH, fuelling NOX and ROS overproduction. Are attracted by lipid peroxidation biomarkers and exacerbate OS. This imbalance favours ROS over antioxidant defences, leading to sperm dysfunction and male infertility. Abbreviations: AC, adenylate cyclase; CAT, catalase; ETC, electron transport chain; G6PD, glucose-6-phosphate dehydrogenase; NO, nitric oxide; NOX, NADPH oxidase; OS, oxidative stress; PKA, protein kinase A; PMNs, polymorphonuclear leukocytes; ROS, reactive oxygen species; SOD, superoxide dismutase.

substances can negatively impact sperm health and male fertility. For example, many people are exposed to heavy metals (cadmium, lead, mercury, arsenic). Exposure occurs through various and distinct routes, including occupational settings, cigarette smoking and contaminated food and water. These heavy metals often exert their toxicity indirectly by interfering with anti-oxidant systems. For instance, they can negatively modulate the uptake and distribution of essential cofactors like Cu and Zn that are important for the functioning of SOD enzymes, a key component of antioxidant defence (Jurasovic et al., 2004). Cd, which is easily absorbed through cigarette smoke, alcohol consumption and contaminated food, can mimic $Ca^{2+}$ in biological reactions. This substitution can disrupt cellular processes, including inhibiting complex II of the ETC in mitochondria, leading to increased production of superoxide anion and high levels of OS (Wang et al., 2004). Bisphenols and phthalates, commonly used in plastic polycarbonate and PVC-based materials to enhance flexibility and elasticity, are another class of ubiquitous environmental toxins that can be released as a result of thermal or mechanical stress and are therefore easily absorbed, making human exposure widespread. Epidemiological evidence suggests that bisphenols and phthalates can negatively impact not only the cardiovascular and neurological systems but also spermatogenesis (Ramadan et al., 2020). Exposure of mice spermatozoa to 100 μM bisphenol A binds to oestrogen receptors at the cytoplasmic (cERs) and nuclear (nErs) levels, altering the spermatozoa proteome and inducing ROS overproduction and lipid peroxidation, ultimately leading to reduced motility, viability and premature acrosomal reaction (Sahu & Jena, 2024). Although these findings provide valuable mechanistic insight, the exposure levels applied are substantially higher than those relevant to human exposure. Phthalates, commonly found in water bottles, food packaging and personal care products, are also a concern because their metabolites were detected in human biological samples, especially in urine, faeces, and blood (Zhang et al., 2021). A study that collected urine samples from 599 IVF (in vitro fertilization) couples identified the presence of eight phthalate metabolites (derived from bis-(2-ethylhexyl) phthalate (DEHP), dibutyl phthalate (DBP) and diethyl phthalate (DEP)) and found a correlation between phthalate metabolite concentrations and markers of OS such as increased CAT activity, MDA levels and 8-OHdG, indicating potential interference with DNA repair mechanisms (Al-Saleh et al., 2019). Microplastics (or small plastic particles) can be found not only in urine but also in human testis and semen (O'Callaghan et al., 2024). These microplastics are associated with reproductive toxicity, especially through the induction of OS and activation of p38 and

Jun N-terminal kinase (JNK)/mitogen-activated protein kinase (MAPK) signalling pathways. Studies in animal models suggest that microplastic exposure can lead to poor sperm quality and reduced testosterone levels (Xie et al., 2020). Taken together, available evidence indicates that exposure to environmental toxins such as heavy metals, bisphenols, phthalates and microplastics can impair male reproductive function primarily through OS–mediated mechanisms. However, most experimental studies focus on exposure levels far exceeding typical human concentrations or are based on animal models, which limits their direct clinical translation in humans. Further large-scale, longitudinal studies are needed to better define dose–response relationships, identify critical exposure windows and assess the combined effects of multiple environmental contaminants on sperm quality and fertility outcomes.

**Psychological stress.** Several reports in the literature have suggested that increased psychological stress is associated with negative effects on sperm parameters and fertility outcomes (vasoconstriction in the testis, anxiety and sexual dysfunction) (Aldhuwayhi et al., 2022). Overall, this leads to an increased level of stress and the release of so-called stress hormones that seem to have a negative impact on sperm parameters and fertility outcomes. Chronic stress increases the circulation of catecholamines such as adrenaline and noradrenaline. These hormones activate the hypothalamic–pituitary–adrenal (HPA) axis by increasing the release of adrenocorticotropic hormone (ACTH). ACTH, in turn, stimulates the production of glucocorticoids, which remain elevated in circulation for 120 min, leading to increased apoptosis in both Sertoli and Leydig cells (Sominsky et al., 2017). Simultaneously, corticotropin-releasing hormone (CRH) inhibits the HPG axis and decreases the release of the hormones LH, follicle-stimulating hormone (FSH) and testosterone, impairing sperm quality and spermatogenesis. Prolonged exposure to excessive stress for long periods can disrupt the body's homeostasis, leading to hormonal and endocrine imbalances, which are often accompanied by OS and inflammation (Barbarino et al., 1989; Ramya et al., 2023). Kolbassi and co-workers demonstrated that mice subjected to general stress for 7 weeks through the 'chronic unpredictable stress' (CUS) protocol, exhibit an increase in corticosterone that results in a strong increase in free radicals, OS and cell membrane destruction, as detected by elevated MDA levels and disrupted blood–testis barrier. Furthermore, in the group subjected to the CUS protocol, corticosterone levels were negatively correlated with testosterone production, reinforcing the detrimental impact of chronic stress on male fertility (Kolbasi et al., 2021). The existing literature shows mixed results, with some studies showing a negative impact of stress on semen parameters and other significant alterations (Hjollund

et al., 2004). This inconsistency may be attributed to the difficulty in conducting human studies on stress due to the number of confounding variables. Moreover, individuals experiencing high stress levels may engage in behaviours known to negatively affect semen quality, such as the use of marijuana, alcohol and other drugs, confounding the results. In conclusion, although the evidence suggests a link between chronic stress and impaired male fertility, particularly through the HPA and HPG axes, further research is necessary to definitively establish the impact of physiological stress on human reproductive health.

**Healthy diet.** Our daily lifestyle influences our fertility, both negatively and positively. Dietary habits play an important role in male reproductive health. Adherence to balanced dietary patterns such as the Mediterranean diet – characterized by the high daily intake of fruits, vegetables, legumes, whole grains, fish and olive oils and a low intake of red meat and processed foods – has been associated with improved semen parameters and antioxidant status (Eslamian et al., 2020; Hosseini et al., 2019). These benefits are likely related to the higher consumption of omega-3 fatty acids, vitamins (A, D, B12 and B2) and polyphenolic compounds, which support membrane integrity and counteract OS (Cristodoro et al., 2024). Several studies demonstrated that individuals with higher fruit and vegetable intake exhibit greater total antioxidant capacity (TAC) and improved sperm motility and viability (Madej et al., 2021). Specific compounds such as resveratrol and epigallocatechin gallate (EGCG) have demonstrated important anti-oxidant, anti-inflammatory and cytoprotective properties in experimental settings, suggesting potential benefits for spermatogenesis and sperm function (Dias et al., 2017; Illiano et al., 2020). Moreover, polyphenols such as anthocyanins can attenuate oxidative damage induced by environmental pollutants by enhancing endogenous antioxidant enzyme systems (CAT, total SOD, GR) and positively modulating many pathways mediated by SIRT1 (Dong et al., 2024). Nonetheless, most available studies are observational, involve limited sample sizes and often lack rigorous control of confounding factors, which restricts the ability to draw firm causal conclusions. Further well-designed, controlled clinical trials are therefore needed to substantiate the observed associations between dietary patterns and fertility. In sum, adherence to a balanced diet – particularly rich in fruits, vegetables and fish, and low in processed foods and red meat – appears to provide essential nutrients and antioxidants that support sperm health, hormone balance and overall reproductive function.

**Regular exercise.** There is evidence that physical exercise can also enhance male reproductive health, particularly sperm quality. Studies in humans demonstrated the beneficial effects of exercise on various sperm parameters. Improvements have been observed in sperm concentration, morphology and progressive motility in sedentary men after a 24-week weightlifting programme (Maleki & Tartibian, 2018) and in men with obesity after a 16-week treadmill running programme (Rosety et al., 2017). These positive effects were associated with increased seminal activity of antioxidant enzymes such as SOD and CAT (Maleki & Tartibian, 2018) while also reducing seminal ROS and pro-inflammatory cytokines (IL-1$\beta$, IL-6, IL-8 and TNF-$\alpha$). Furthermore, exercise has been linked to increased serum sex hormone binding globulin (SHBG) and testosterone levels in men with obesity after a 14-week exercise programme (Håkonsen et al., 2011). Rodent studies showed similar results, including increased sperm concentration in C57BL/6 mice after a 21-week treadmill programme (Xu et al., 2022) and in Sprague–Dawley rats after a 6-week swimming programme (Elmas et al., 2019). Motility improvements have also been documented in C57BL/6 mice subjected to treadmill exercise for 8 (Nematollahi et al., 2019) or 12 weeks (Xu et al., 2022), and improvements in sperm morphology were also reported in Sprague–Dawley rats after 6 weeks of swimming (Elmas et al., 2019). These beneficial effects in animal models were accompanied by increased testosterone, FSH and LH levels, along with decreased leptin levels in Sprague–Dawley rats after a 6-week swimming programme (Elmas et al., 2022) when compared to animals with obesity. Reductions in oxidative damage, particularly lipid peroxidation and DNA damage, have also been observed in exercised animals (Nematollahi et al., 2019). The positive influence of lifelong physical activity extends to testicular somatic cells, such as Sertoli and Leydig cells. Studies have shown increased populations of these cells in C57BL/6 mice after 14 months of running wheel activity compared to sedentary animals (Chigurupati et al., 2008). Exercise, particularly of moderate intensity, has a positive impact on spermatogenesis and spermiogenesis, particularly under moderate-intensity conditions. Moderate swimming in Sprague–Dawley rats for 6 weeks (Elmas et al., 2022) or moderate treadmill exercise in Wistar rats for 12 weeks (Azar et al., 2020) has been shown to enhance spermatogenesis and spermiogenesis compared to obese controls, likely due to elevated serum FSH levels (observed in mice (Xu et al., 2022) and in rats after a treadmill (Karaman & Tektemur, 2022) or swimming regimen (Elmas et al., 2022)). Beyond increased numbers, Sertoli cells (SCs) also exhibit enhanced functionality, including higher LDH activity, upregulated glucose transporters (GLUT-1 and GLUT-3) and increased expression of Igf1 and monocarboxylate transporter 4 (MCT-4), which collectively support lactate production and germ cell sub-

strate uptake (Maleki et al., 2024). Steroidogenesis, like spermatogenesis, is positively impacted by physical activity and negatively affected by obesity. This improvement is likely driven by increased serum LH levels (Elmas et al., 2022) and Leydig cell numbers. Rodent studies involving swimming or treadmill exercise demonstrated increased Leydig cell function reflected in the increased expression of steroidogenic markers such as steroidogenic acute regulatory protein (StAR), cytochrome P450 side chain cleavage enzyme (CYP11A1), P450c17a (17-$\alpha$-hydroxylase) and 17-lyase (CYP17A1) and steroidogenic factor 1 (SF-1) compared to obese controls (Yi et al., 2017). In humans, moderate-intensity weightlifting improves testicular function by reducing seminal inflammation (IL-1$\beta$, IL-6, IL-8, TNF-$\alpha$) and OS (ROS, lipid peroxidation), simultaneously enhancing antioxidant defences (SOD, CAT), leading to better sperm quality (Maleki & Tartibian, 2018). Conversely, high-intensity exercise, such as triathlon, can hamper these benefits and even increase testicular macrophage infiltration (Vaamonde et al., 2018). In rodents, moderate exercise enhances reproductive health by improving Sertoli cell function (Maleki et al., 2024), increasing testosterone level (Yi et al., 2017), enhancing antioxidant defences (Yi et al., 2020) and reducing germ cell DNA damage (Azar et al., 2020). In contrast, high-intensity exercise induces OS, inflammation and testicular dysfunction, mimicking obesity-like effects, including reduced sperm motility and lower testosterone levels, and steroidogenic enzyme activity (Yi et al., 2020). One of the most prominent mechanisms by which physical exercise seems to regulate antioxidant defences and exert positive effects on the male reproductive system is the SIRT1–Nrf2 pathway. This pathway is compromised in obesity due to miR-34a overexpression, which inhibits the expression of SIRT1 (Zhu et al., 2021). SIRT1, which activates Nrf2 by deacetylation, is overexpressed in Wistar rats after 10 weeks of running wheel physical activity (Heydari et al., 2021). This leads to Nrf2 activation and subsequent upregulation of antioxidant enzymes such as heme oxygenase-1 (HO-1) and NAD(P)H:quinone oxidoreductase (NQO) (Xu et al., 2022) in both germ cells and Leydig cells, favouring spermatogenesis (Coussens et al., 2008) and steroidogenesis (Chung et al., 2021), respectively. Overall, the positive effects exercise exerts on the reproductive axis encompass improvement in sperm parameters, improvement in Leydig and Sertoli cell activity and an overall reduction in OS and inflammation. Whereas moderate exercise offers consistent reproductive health benefits, high-intensity regimens may pose risks and require further study to optimize outcomes. The SIRT1–Nrf2 axis emerges as a promising target to modulate testicular OS and mitigate damage induced by lifestyle factors, such as diets rich in fats.

## Conclusions and future perspectives

The discussion highlights how OS plays a dual and crucial role in the physiology and pathology of spermatogenesis. Although ROS are essential for specific processes such as capacitation and hyperactivation, their imbalance and excess can severely compromise sperm quality through DNA alterations, lipid peroxidation and reduced motility. This fine balance makes it difficult to define a clear clinical demarcation between physiological and pathological ROS levels. The evidence discussed underscores how this equilibrium is influenced by several lifestyle and environmental factors, including pollution, smoking, diet and physical activity. Tobacco smoke, for instance, combines heavy metal exposure with the toxic effects of cotinine and other compounds, synergistically impairing spermatogenesis. Although complete avoidance of heavy metals is difficult, studies suggest that smoking cessation alone can significantly reduce serum levels and partially restore sperm quality (Adams & Newcomb, 2014). Dietary patterns also modulate redox homeostasis. The Mediterranean diet, rich in omega-3 fatty acids, vitamins and minerals, supports antioxidant defence and sperm function. Polyphenols such as resveratrol and flavonoids exhibit potential in mitigating OS and improving sperm parameters, although their poor oral bioavailability remains a major limitation, calling for new delivery strategies such as nanoformulations or targeted supplements. Beyond dietary improvements and supplementation, physical exercise stands as a lifestyle factor to mitigate OS in the male reproductive tract. As explained, regular physical exercise can enhance positive effects on the metabolic and reproductive health of men, although this effect depends on the intensity of the exercise regimen. For sedentary infertile men at baseline, moderate physical exercise reduces seminal inflammatory cytokines ROS and OS, whereas it increases the activity of antioxidant enzymes SOD and CAT. This is reflected in improved sperm parameters, such as progressive motility, normal morphology and concentration (Moreira et al., 2025). Overall, integrating lifestyle modifications – smoking cessation, balanced diet and regular exercise – represents a practical and evidence-based approach to counteract OS in male reproduction. Nonetheless, most human studies remain correlative, emphasizing the need for well-controlled clinical trials and mechanistic investigations to better define the causal links between lifestyle factors, OS and male infertility.

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

## Additional information

### Competing interests

All authors declare that there are no competing interests and that they also have no conflict of interest in accordance with journal policy to disclose.

## Author contributions

R.J.M.: conceptualization, methodology, writing – review and editing. A.R.: validation, writing - review and editing. M.L.P.: conceptualization, methodology, writing – original draft, writing – review and editing. D.B.: conceptualization, methodology, writing – original draft, writing – review and editing. P.F.O.: conceptualization, methodology, writing – original draft, writing – review and editing, funding acquisition. M.G.A.: conceptualization, supervision, methodology, writing – original draft, writing – review and editing, funding acquisition.

## Funding

P.F.O. is supported by national funds through FCT – Fundação para a Ciência e a Tecnologia, I.P., under the Scientific Employment Stimulus-Institutional Call-Reference CEEC-INST/00026/2018. This work also received support from FCT/MCTES to LAQV-REQUIMTE (LA/P/0008/202 - DOI 10.54499/LA/P/0008/2020; UIDP/50006/2020 - DOI 10.54499/UIDP/50006/2020; and UIDB/50006/2020 - DOI 10.54499/UIDB/50006/2020) and to iBiMed (UIDB/04501/2020 - DOI 10.54499/UIDB/04501/2020 and UIDP/04501/2020 - DOI 10.54499/UIDP/04501/2020), through national funds. M.L.P. is funded by project CICECO – Aveiro Institute of Materials, UID/50011/2025 (DOI 10.54499/UID/50011/2025) and LA/P/0006/2020 (DOI 10.54499/LA/P/0006/2020), financed by national funds through the FCT/MCTES (PIDDAC).

## Acknowledgements

The authors acknowledge past and present co-workers for their contributions to this work.

Open access publication funding provided by FCT (b-on).

## Keywords

lifestyle, metabolism, oxidative stress, sperm, spermatogenesis

## Supporting information

Additional supporting information can be found online in the Supporting Information section at the end of the HTML view of the article. Supporting information files available:

**Peer Review History**

