## [Peer Review History · The Journal of Physiology]

Lifestyle Implications to the Paradox and Managing of Oxidative Stress in Sperm

Giuseppe Tancredi Patané, Ruben J. Moreira, Annamaria Russo, Maria Lourdes Pereira, Pedro F Oliveira, Davide Barreca, and Marco G. Alves

DOI: 10.1113/JP289694

Corresponding author(s): Marco Alves (marcoalves@ua.pt)

Review Timeline:	Submission Date:	07-Aug-2025
	Editorial Decision:	06-Oct-2025
	Revision Received:	30-Oct-2025
	Editorial Decision:	03-Nov-2025
	Revision Received:	16-Nov-2025
	Accepted:	19-Nov-2025

Senior Editor: Laura Bennet

Reviewing Editor: Rebecca Simmons

Transaction Report:

Dear Professor Alves,

Re: JP-TR-2025-289694 "Lifestyle Implications to the Paradox and Managing of Oxidative Stress in Sperm" by Giuseppe Patané, Ruben moreira, Anna Annamaria Russo, Maria Lourdes Pereira, Pedro F Oliveira, Davide Barreca, and Marco Alves

Thank you for submitting your manuscript to The Journal of Physiology. It has been assessed by a Reviewing Editor and by 2 expert referees and we are pleased to tell you that it is acceptable for publication following satisfactory revision.

ABSTRACT FIGURES: Authors may use The Journal's premium BioRender account to create/redraw their Abstract Figures (and any other suitable schematic figure). Information on how to access this account is here: <https://physoc.onlinelibrary.wiley.com/journal/14697793/biorender-access>.

REVISION CHECKLIST: Upload a full Response to Referees file. To create your 'Response to Referees' copy all the reports, including any comments from the Senior and Reviewing Editors, into a Microsoft Word, or similar, file and respond to each point, using font or background colour to distinguish comments and responses and upload as the required file type.

We look forward to receiving your revised submission.

Yours sincerely,

Laura Bennet

REQUIRED ITEMS

- Please include an Abstract Figure file, as well as the Figure Legend text within the main article file. The Abstract Figure is a piece of artwork designed to give readers an immediate understanding of the Review Article and should summarise the main conclusions. If possible, the image should be easily 'readable' from left to right or top to bottom. It should show the physiological relevance of the Review so readers can assess the importance and content of the article. Abstract Figures should not merely recapitulate other figures in the Review. Please try to keep the diagram as simple as possible and without superfluous information that may distract from the main conclusion of the Review. Abstract Figures must be provided by authors no later than the revised manuscript stage and should be uploaded as a separate file during online submission labelled as File Type 'Abstract Figure'. Please ensure that you include the figure legend in the main article file. All Abstract Figures will be sent to a professional illustrator for redrawing and you may be asked to approve the redrawn figure before your paper is accepted.

- Your MS must include a complete "Additional information section" with the following 4 headings and content:

Competing Interests: A statement regarding competing interests. If there are no competing interests, a statement to this effect must be included. All authors should disclose any conflict of interest in accordance with journal policy.

Author contributions: Each author should take responsibility for a particular section of the study and have contributed to writing the paper. Acquisition of funding, administrative support or the collection of data alone does not justify authorship; these contributions to the study should be listed in the Acknowledgements. Additional information such as 'X and Y have contributed equally to this work' may be added as a footnote on the title page.

It must be stated that all authors approved the final version of the manuscript and that all persons designated as authors qualify for authorship, and all those who qualify for authorship are listed.

Funding: Authors must indicate all sources of funding, including grant numbers. If authors have not received funding, this must be stated.

It is the responsibility of authors funded by RCUK to adhere to their policy regarding funding sources and underlying research material. The policy requires funding information to be included within the acknowledgement section of a paper. Guidance on how to acknowledge funding information is provided by the Research Information Network. The policy also requires all research papers, if applicable, to include a statement on how any underlying research materials, such as data, samples or models, can be accessed. However, the policy does not require that the data must be made open. If there are considered to be good or compelling reasons to protect access to the data, for example commercial confidentiality or legitimate sensitivities around data derived from potentially identifiable human participants, these should be included in the statement.

Acknowledgements: Acknowledgements should be the minimum consistent with courtesy. The wording of acknowledgements of scientific assistance or advice must have been seen and approved by the persons concerned. This section should not include details of funding.

- The reference list must be in alphabetical order, rather than numbered, to comply with our Journal format.

- Author profile(s) must be uploaded via the submission form. Authors should submit a short biography (no more than 100 words for one author or 150 words in total for two authors) and a portrait photograph of the two leading authors on the paper. These should be uploaded and clearly labelled together in a Word document with the revised version of the manuscript. Any standard image format for the photograph is acceptable, but the resolution should be at least 300 DPI and preferably more. A group photograph of all authors is also acceptable, providing the biography for the whole group does not exceed 150 words.

EDITOR COMMENTS

Reviewing Editor:

Thank you for submitting your review. The second reviewer had several suggestions for improvement. Accordingly, we invite you to revise your manuscript and resubmit.

Please also see 'Required Items' above.

REFeree COMMENTS

Referee #2:

This is a well written comprehensive review dealing with the role of ROS in sperm functions. I have no further comments.

Referee #3:

The review is comprehensive and generally well written. I have a several suggestions to improve the clarity of the review.

1. Importantly, the review should address some of the weaknesses of and the gaps in the studies that were mentioned in all sections of the review:

a. In the Diet section, a study was discussed that showed a correlation with high levels of SFA intake and reduced sperm concentration and motility. The authors state in Lines 178/179, "Thus, a correlation between those two facts can be established." This conclusion should be rewritten to reflect an association, and the authors should comment on the weaknesses and strengths of this study (and others throughout the review).

b. Another example is the study by Oda and El-Maddawy that used very high levels of deltamethrin, making extrapolation to the human difficult.

c. Similarly, the authors should discuss the relevance to humans of the study in lines 403-404 showing high levels of OS cause sperm in humans. What clinical situations can cause such high levels of OS?

d. The study by Sahu and Jena (lines 429-432) used extraordinarily high levels of BPA that are not clinically relevant. This should be discussed as a limitation to the study.

e. I suggest leaving out the EMR study. This is quite controversial, and few studies support the conclusion.

f. The conclusion sentence in the Environmental toxins should be expanded to include limitations of the studies and gaps in the literature as noted above for other sections.

g. I suggest removing the first several sentences in the Psychological stress section (452-459) and start the section with a general statement about reports in the literature which have suggested that increased stress is associated with negative impacts on sperm parameters and fertility outcomes. The discussion on the controversy is excellent, however, there are several typos: line 473-should be mixed, and highlighting should be changed to showing a negative impact. I suggest that lines 475/476 should be rewritten: " This inconsistency may be attributed to the difficulty of conducting human studies on stress due to the number of confounding variables". I would remove the sentence "A reference control without stress..." I also suggest removing the word strong at the end of line 478 as the studies are not robust.

h. The Healthy Diet section should be significantly shortened and rewritten or possibly removed. I suggest removing lines 483-490 as they are not germane to the discussion. Most of the cited studies are observational, not well controlled, with small numbers of participants. The last sentence of this section should be removed as it is not relevant to a scientific review.

i. I suggest adding a paragraph at the end of each section explaining that most of these studies in humans are correlative. Studies in animals do provide some support, and this should be discussed in the context of human studies when appropriate. The alcohol section is excellent and does discuss the limitations of studies linking alcohol to sperm DNA fragmentation. This should be done throughout the review for the other exposures.

- j. The first sentence of the Exercise section should be removed as it is not germane to the review.
- k. I suggest removing the Healthy Weight Section. There are very few studies on this topic with regard to sperm parameters. While it is conceivable that weight loss, weight regain cycles could impact sperm parameters, without sufficient evidence, it is not helpful to include in the Review.
- l. I also suggest leaving out the stress management strategies as the beneficial effects have not been rigorously studied.
2. The first paragraph is quite long, and I suggest breaking it up into more than one paragraph.
3. The conclusion section is overly long and should be significantly shortened to highlight the important parts of the review and not just restate what was written above.
4. The study cited in lines 440-441 needs a citation.
5. The Omolaoye et al citation is a review. The concept that cigarette smoke has a transgenerational effect in mice is controversial. The authors should cite the original paper and describe the strengths and weaknesses of this original study. Further, the statement in lines 242-243 (the potential health of future generations) is too strong. I suggest omitting it.
6. I suggest omitting the sentence in lines 245 and changing it to "Alcohol has been shown in human and animal studies to affect spermatogenesis.."
7. I suggest omitting the Disease section. There are numerous diseases that affect sperm quality and only highlighting a few does not make sense in the context of lifestyle and environmental factors in this review.
8. Are the blood levels of carotenoid being routinely used as a biomarker for sperm quality? The cited study is over 10 years old and if this not being routinely done, I suggest either stating that or omitting this.
9. The section on Physiological roles for ROS is excellent.
10. There are several typographical errors:
 - a. Line 117: change the word on to "to"
 - b. Line 174: change the word levels to level
 - c. Lines 133 and 261 are not grammatically correct.
 - d. Suggest leaving out the word "should" in line 263 and changing deserve to deserves.
 - e. Line 428: remove the word with at the end of the line
 - f. There is an extra space between the end of the word and the period in lines: 123,128, 307, 317, 324, 330,
11. In line 211, I suggest changing the word ambivalent to contradictory

END OF COMMENTS

Aveiro, 30th October 2025

Dear Editor,

We thank you for allowing us to revise our manuscript entitled “**Lifestyle Implications to the Paradox and Managing of Oxidative Stress in Sperm**” by Patanè et al. (JP-TR-2025-289694). We acknowledge the Reviewers for all their valuable comments, constructive criticism, and suggestions that permitted us to improve the scientific quality of the manuscript. All suggestions were considered, and modifications were made accordingly.

We hope that all the changes we have included in this revised version, have improved the quality of our manuscript and that it is now suitable for publication in ***Journal of Physiology***. Please find enclosed the answers to all the queries and the revised version of our manuscript.

With our best regards,

On behalf of all the co-authors,

Marco G. Alves, PhD

Principal Investigator
Department of Medical Sciences, Institute of Biomedicine (iBiMED)
University of Aveiro, Aveiro, Portugal
e-mail: alvesmarc@gmail.com

Editor #1: *"Required Items"*.

Answer: Dear editor, thank you for your suggestions. In accordance with this, we added all the required items that you suggested. You can find attached the abstract figure, and in the text also the abstract figure legend. Then we also added the other required items (conflict of interest, author contributions..etc).

Reviewer #2:

We thank Reviewer #2 for the positive feedback.

Title This is a well written comprehensive review dealing with the role of ROS in sperm functions. I have no further comments.

Answer: Thank you for your positive comment.

Reviewer #3:

We thank Reviewer #3 for the positive feedback and suggestions that allow us to improve the quality of the review.

1. a). In the Diet section, a study was discussed that showed a correlation with high levels of SFA intake and reduced sperm concentration and motility. The authors state in Lines 178/179, "Thus, a correlation between those two facts can be established." This conclusion should be rewritten to reflect an association, and the authors should comment on the weaknesses and strengths of this study (and others throughout the review).

Answer: Thank you for your suggestion. We remove the sentence "Thus, a correlation between those two facts can be established" from this part and suggest the following "Therefore, these findings suggest an association between high SFA intake and impaired sperm parameters. However, due to the observational nature of the available studies and the potential presence of confounding factors, a precise correlation cannot be definitively established."

1. b). *Another example is the study by Oda and El-Maddawy that used very high levels of deltamethrin, making extrapolation to the human difficult.*

Answer: Thank you for your suggestion. We added the sentence "However, the doses of deltamethrin used in this *in vivo* study were markedly higher than typical human exposure levels, which limits the direct translation of these findings to humans."

1. c): *Similarly, the authors should discuss the relevance to humans of the study in lines 403-404 showing high levels of OS cause sperm in humans. What clinical situations can cause such high level of OS?*

Answer: Thank you for your suggestion. In accordance with this we changed the end of this section and we added an explanation for high level of OS " In conclusion, while ROS play a vital signaling role in sperm physiology at low concentrations, OS, resulting from an imbalance between ROS production and antioxidant capacity, severely compromises sperm quality and fertility. By damaging sperm membranes and DNA, OS interferes with key processes such as capacitation, hyperactivation, and the acrosome reaction, ultimately hindering successful fertilization. (Figure 3). In humans, elevated OS levels are observed in various clinical and environmental conditions- including varicocele, genital tract infections, inflammation, obesity, smoking, and exposure to environmental pollutants or heat stress. These situations can significantly increase ROS generation beyond the antioxidant capacity of seminal plasma, thereby mirroring the high OS levels reported in experimental studies. Therefore, maintaining redox homeostasis is essential for optimal sperm function and male reproductive health.

1. d). *The study by Sahu and Jena (lines 429-432) used extraordinarily high levels of BPA that are not clinically relevant. This should be discussed as a limitation to the study.*

Answer: Thank you for your suggestion. In accordance with this, we added the sentence “ While these findings provide valuable mechanistic insight, the exposure levels applied are substantially higher than those relevant to human exposure.”

(1. e). I suggest leaving out the EMR study. This is quite controversial, and few studies support the conclusion.

Answer: Thank you for your suggestion. In accordance with this we removed this part from the section: “ How lifestyle impacts the paradox of oxidative stress in sperm physiology and male fertility: 1) Environmental toxins”

(1. f). The conclusion sentence in the Environmental toxins should be expanded to include limitations of the studies and gaps in the literature as noted above for other sections.

Answer: Thank you for your suggestion. In accordance with this, we added this sentence to highlights the limit of some reported studies: “Taken together, available evidence indicates that exposure to environmental toxins such as heavy metals, bisphenols, phthalates, and microplastics can impair male reproductive function primarily through oxidative stress-mediated mechanisms. However, most experimental studies focus on exposure levels far exceeding typical human concentrations, or are based on animal models, which limits their direct clinical translation in humans. Further large-scale, longitudinal studies are needed to better define dose-response relationships, identify critical exposure windows, and assess the combined effects of multiple environmental contaminants on sperm quality and fertility outcomes”.

1. g). *I suggest removing the first several sentences in the Psychological stress section (452-459) and start the section with a general statement about reports in the literature which have suggested that increased stress is associated with negative impacts on sperm parameters and fertility outcomes. The discussion on the controversy is excellent, however, there are several typos: line 473-should be mixed, and highlighting should be changed to showing a negative impact. I suggest that lines 475/476 should be rewritten: " This inconsistency may be attributed to the difficulty of conducting human studies on stress due to the number of confounding variables". I would remove the sentence "A reference control without stress..." I also suggest removing the word strong at the end of line 478 as the studies are not robust.*

Answer: Thank you for your suggestion. In accordance with this, we changed the sentence in the introductory sentences in this section. In particular, we started directly in this way: " Several reports in the literature have suggested that increased psychological stress is associated with negative effects on sperm parameters and fertility outcomes (vasoconstriction in the testis, anxiety and sexual dysfunction)". Then, as you suggested in the comment, we changed the lines 475/476. Finally, we removed the word "strong" in the last sentence.

1. h). *The Healthy Diet section should be significantly shortened and rewritten or possibly removed. I suggest removing lines 483-490 as they are not germane to the discussion. Most of the cited studies are observational, not well controlled, with small numbers of participants. The last sentence of this section should be removed as it is not relevant to a scientific review.*

Answer: Thank you for your suggestion. We have therefore significantly shortened and rewritten this section, removing the introductory historical context (lines 483–490) and the final non-scientific sentence. The revised version now focuses on key findings from the literature regarding dietary patterns and male fertility, while acknowledging the limitations of existing studies, which are mostly observational and based on small cohorts. In detail, here the

new version of this sections: Our daily lifestyle influences our fertility, both negatively and positively. Dietary habits play an important role in male reproductive health. Adherence to balanced dietary patterns such as the Mediterranean diet- characterized by the high daily intake of fruits, vegetables, legumes, whole grains, fish and olive oils and a low intake of red meat and processed foods- has been associated with improved semen parameters and antioxidant status (Hosseini *et al.*, 2019; Eslamian *et al.*, 2020). . These benefits are likely related to the higher consumption of omega-3, fatty acids, vitamins (A, D, B12 and B2) and polyphenolic compounds, which support membrane integrity and counteract oxidative stress. (Cristodoro *et al.*, 2024). Several studies demonstrated that individuals with higher fruit and vegetable intake exhibit greater total antioxidant capacity (TAC) and improved sperm motility and viability (Madej *et al.*, 2021). Specific compounds such as resveratrol and epigallocatechin gallate (EGCG) have demonstrated important antioxidants, anti-inflammatory and cytoprotective properties in experimental settings, suggesting potential benefits for spermatogenesis and sperm function (Illiano *et al.*, 2020)(Dias *et al.*, 2017). Moreover, polyphenols such as anthocyanins can attenuate oxidative damage induced by environmental pollutants by enhancing endogenous antioxidant enzyme systems (CAT, total SOD, GR) and the positive modulation on many pathways mediated by SIRT1 (Dong *et al.*, 2024). Nonetheless, most available studies are observational, involve limited samples sizes, and often lack rigorous control of confounding factors, which restricts the ability to draw firm causal conclusions. Further well-designed, controlled clinical trials are therefore needed to substantiate the observed associations between dietary patterns and fertility. In sum, adherence to a balanced diet- particularly rich in fruits, vegetables and fish, and low in processed foods and red meat- appears to provide essential nutrients and antioxidants that support sperm health, hormone balance, and overall reproductive function.”

1. i). I suggest adding a paragraph at the end of each section explaining that most of these studies in humans are correlative. Studies in animals do provide some support, and this should be discussed in the context of human studies when appropriate. The alcohol section is excellent and does discuss the limitations of studies linking alcohol to sperm DNA fragmentation. This should be done throughout the review for the other exposures

Answer: Thank you for your suggestion. We tried to add “last sentences” to emphasize that most available human studies are observational and that evidence from animal studies, while mechanistically supportive, should be interpreted with caution when extrapolating to humans. These additions ensure greater consistency across the review and align the discussion of each exposure with the approach used in the alcohol section.

1. j). The first sentence of the Exercise section should be removed as it is not germane to the review.

Answer: Thank you for your suggestion. In accordance with this, we removed the first sentence in the Exercise section.

1. k). I suggest removing the Healthy Weight Section. There are very few studies on this topic with regard to sperm parameters. While it is conceivable that weight loss, weight regain cycles could impact sperm parameters, without sufficient evidence, it is not helpful to include in the Review.

Answer: Thank you for your suggestion. In accordance with this, we removed the Healthy weight sections.

1. 1). *I also suggest leaving out the stress management strategies as the beneficial effects have not been rigorously studied.*

Answer: Thank you for your suggestion. In accordance with this, we removed the “Stress management strategies section”.

2). *The first paragraph is quite long, and I suggest breaking it up into more than one paragraph.*

Answer: Thank you for your suggestion. In accordance with this, we reduced the introduction.

3). *The conclusion section is overly long and should be significantly shortened to highlight the important parts of the review and not just restate what was written above*

Answer: Thank you for your suggestions. We tried to reduce the conclusion section and to highlight the important parts of the review. In detail “ The discussion highlights how OS plays a dual and crucial role in the physiology and pathology of spermatogenesis. While ROS are essential for specific processes such as capacitation and hyperactivation, their imbalance and excess can severely compromise sperm quality through DNA alterations, lipid peroxidation and reduced motility. This fine balance makes it difficult to define a clear clinical demarcation between physiological and pathological ROS levels. The evidence discussed underscores how this equilibrium is influenced by several lifestyle and environmental factors, including pollution, smoking, diet, and physical activity. Tobacco smoke, for instance, combines heavy metal exposure with the toxic effects of conitine and other compounds, synergistically impairing spermatogenesis. Although complete avoidance of heavy metals is difficult, studies suggest that smoking cessation alone can significantly reduce serum levels and partially restore sperm quality (Adams & Newcomb, 2014). Dietary patterns also modulate redox homeostasis. The mediterranean diet, rich in omega-3 fatty acids, vitamins, and minerals, supports

antioxidant defense and sperm function. Polyphenols such as resveratrol and flavonoids show potential in mitigating OS and improving sperm parameters, although their poor oral bioavailability remains a major limitation, calling for new delivery strategies such as nanoformulations or targeted supplements. Beyond dietary improvements and supplementation, physical exercise stands as a lifestyle factor to mitigate OS in male reproductive tract. As explained, regular physical exercise can boost positive effects on the metabolic and reproductive health of men, albeit this effect depends on the intensity of the exercise regimen. For sedentary infertile men at baseline, moderate physical exercise reduces seminal inflammatory cytokines, ROS and OS, while increasing the activity of antioxidant enzymes SOD and CAT. This is reflected in improved sperm parameters, such as progressive motility, normal morphology and concentration (Moreira *et al.*, 2025). Overall, integrating lifestyle modifications—smoking cessation, balanced diet, and regular exercise—represents a practical and evidence-based approach to counteract oxidative stress in male reproduction. Nonetheless, most human studies remain correlative, emphasizing the need for well-controlled clinical trials and mechanistic investigations to better define the causal links between lifestyle factors, OS, and male infertility.”

(4). The study cited in lines 440-441 needs a citation.

Answer: Thank you for your suggestion. As you can see, we added the missed citation.

5). The Omolaoye et al citation is a review. The concept that cigarette smoke has a transgenerational effect in mice is controversial. The authors should cite the original paper and describe the strengths and weaknesses of this original study. Further, the statement in lines 242-243 (the potential health of future generations) is too strong. I suggest omitting it.

Answer: Thank you for your suggestions. We added new citations that refers to an original article. In accordance, we also changed the lines 242-243 eliminating “the potential health of future generations”.

6). *I suggest omitting the sentence in lines 245 and changing it to "Alcohol has been shown in human and animal studies to affect spermatogenesis."*

Answer: Thank you for your suggestion. We changed the first sentence as you said.

7). *I suggest omitting the Disease section. There are numerous diseases that affect sperm quality and only highlighting a few does not make sense in the context of lifestyle and environmental factors in this review.*

Answer: Thank you for your suggestion. In accordance with this we removed the disease section.

8). *Are the blood levels of carotenoid being routinely used as a biomarker for sperm quality? The cited study is over 10 years old and if this not being routinely done, I suggest either stating that or omitting this.*

Answer: Thank you for your suggestion. In accordance with this we changed the period: “Clinical studies have further demonstrated that blood carotenoid levels may serve as potential biochemical biomarkers for predicting sperm quality, OS, and ROS levels in sperm cells (Benedetti *et al.*, 2012). However, this approach remains largely experimental and is not routinely applied in clinical settings.”

9). *The section on Physiological roles for ROS is excellent.*

Answer: Thank you.

10. a,b) There are several typographical errors: a. Line 117: change the word on to "to" of, b.

Line 174: change the word levels to level.

Answer: Thank you for your suggestions. We corrected these sentences.

10. c). Lines 133 and 261 are not grammatically correct.

Answer: Thank you for your suggestions. We changed both the sentences. In detail for 133 line: "A correlation between sperm morphology and ROS production was first described in 1996, with morphologically normal spermatozoa exhibiting lower production of free radicals. More recently, the term Sperm Deformity Index (SDI) was introduced, which quantitatively represents the number of sperm deformities and thus indirectly predicts ROS production in sperm with borderline morphology (e.g. acrosomal damage, cytoplasmic droplets, amorphous heads) (Walczak-Jedrzejowska *et al.*, 2013)". For line 261: " Overall, alcohol consumption remains a threat to male reproductive health by affecting spermatogenesis, hormone balance and even the function of testicular cells. The detrimental effects of chronic alcohol consumption on sperm parameters (e.g. volume, morphology) are well established although whether OS is the underlying link remains a matter of debate."

10. d,e,f): Suggest leaving out the word "should" in line 263 and changing deserve to deserves,

Line 428: remove the word with at the end of the line, There is an extra space between the end of the word and the period in lines: 123,128, 307, 317, 324, 330.

Answer: Thank you for your suggestions. We do all the changes and we eliminated the extra spaces.

11). *In line 211, I suggest changing the word ambivalent to contradictory*

Answer: Thank you for your suggestion. Done.

We hope that all the changes we have included in this revised version, have improved the quality of our manuscript and that it is now suitable for publication in ***Journal of Physiology***. Please find enclosed the answers to all the queries and the revised version of our manuscript.

With our best regards,

On behalf of all the co-authors,

Marco G. Alves, PhD

Principal Investigator
Department of Medical Sciences, Institute of Biomedicine (iBiMED)
University of Aveiro, Aveiro, Portugal
e-mail: alvesmarc@gmail.com

Dear Professor Alves,

Re: JP-TR-2025-289694R1 "Lifestyle Implications to the Paradox and Managing of Oxidative Stress in Sperm" by Giuseppe Tancredi Patané, Ruben moreira, Annamaria Russo, Maria Lourdes Pereira, Pedro F Oliveira, Davide Barreca, and Marco Alves

Thank you for submitting your manuscript to The Journal of Physiology. It has been assessed by a Reviewing Editor and by 1 expert referee and we are pleased to tell you that it is acceptable for publication following satisfactory revision.

ABSTRACT FIGURES: Authors may use The Journal's premium BioRender account to create/redraw their Abstract Figures (and any other suitable schematic figure). Information on how to access this account is here: <https://physoc.onlinelibrary.wiley.com/journal/14697793/biorender-access>.

REVISION CHECKLIST: Upload a full Response to Referees file. To create your 'Response to Referees' copy all the reports, including any comments from the Senior and Reviewing Editors, into a Microsoft Word, or similar, file and respond to each point, using font or background colour to distinguish comments and responses and upload as the required file type.

We look forward to receiving your revised submission.

Yours sincerely,

Laura Bennet

REQUIRED ITEMS

1)- Please provide higher quality and higher resolution images and figures.

2) - If applicable, it is the authors' responsibility to obtain any necessary permissions to reproduce previously published material and to list these within the main article file. For information, please see: https://jp.msubmit.net/cgi-bin/main.plex?form_type=display_requirements#permissions.

3) - Please include a full title page as part of your main article (Word) file, which should contain the following: title, authors, affiliations, corresponding author name and contact details, keywords, and running title.

4) - The corresponding author must provide an institutional email address (not a personal address) for their author account. We encourage ALL co-authors to also provide institutional email addresses. If this cannot be provided (as corresponding author), then a stamped letter must be provided from the institution which confirms their role and employment there (please upload this with the revised submission).

EDITOR COMMENTS

Reviewing Editor:

Thank you for responding to the reviewers' suggestions.

REFEREE COMMENTS

Referee #3:

Thank you for addressing my concerns and incorporating my suggestions. I believe this has strengthened the review and that readers will find it interesting.

END OF COMMENTS

Aveiro, 11th November 2025

Dear Editor,

We are grateful that our manuscript entitled "**Lifestyle Implications to the Paradox and Managing of Oxidative Stress in Sperm**" by Patanè et al. (JP-TR-2025-289694R1) is considered suitable for publication in the Journal of Physiology. We sincerely appreciate the constructive comments, criticism, and valuable suggestions provided by the Reviewers, which have truly helped to improve the scientific quality and clarity of our work. All suggestions were carefully considered and the manuscript has been revised accordingly. We believe the changes we have incorporated have enhanced our submission, and we thank you again for the opportunity to contribute to your journal.

Please find enclosed our responses to all Reviewer queries, along with the revised version of the manuscript.

With our best regards,

On behalf of all the co-authors,

Marco G. Alves, PhD

Principal Investigator
Department of Medical Sciences, Institute of Biomedicine (iBiMED)
University of Aveiro, Aveiro, Portugal
e-mail: alvesmarc@gmail.com

Editor #1: Title "Required Items".

1)- Please provide higher quality and higher resolution images and figures.

Answer: Dear Editor, thank you for your suggestion. We have improved the quality and resolution of the images, making them more suitable for publication.

2) - If applicable, it is the authors' responsibility to obtain any necessary permissions to reproduce previously published material and to list these within the main article file. For information, please see: https://jp.msubmit.net/cgi-bin/main.plex?form_type=display_requirements#permissions.

Answer: All images used in this publication have been originally created by us and are used solely for this review.

3) - Please include a full title page as part of your main article (Word) file, which should contain the following: title, authors, affiliations, corresponding author name and contact details, keywords, and running title.

Answer: Dear editor, it's already done.

4) - The corresponding author must provide an institutional email address (not a personal address) for their author account. We encourage ALL co-authors to also provide institutional email addresses. If this cannot be provided (as corresponding author), then a stamped letter must be provided from the institution which confirms their role and employment there (please upload this with the revised submission).

Answer: Dear editor, in the first page the corresponding author (Marco G. Alves) used his institutional email address (marcoalves@ua.pt)

Dear Professor Alves,

Re: JP-TR-2025-289694R2 "Lifestyle Implications to the Paradox and Managing of Oxidative Stress in Sperm" by Giuseppe Tancredi Patané, Ruben J. Moreira, Annamaria Russo, Maria Lourdes Pereira, Pedro F Oliveira, Davide Barreca, and Marco G. Alves

We are pleased to tell you that your paper has been accepted for publication in The Journal of Physiology.

Authors should note that it is too late at this point to offer corrections prior to proofing. Major corrections at proof stage, such as changes to figures, will be referred to the Editors for approval before they can be incorporated. Only minor changes, such as to style and consistency, should be made at proof stage. Changes that need to be made after proof stage will usually require a formal correction notice.

Yours sincerely,

Laura Bennet
Senior Editor
The Journal of Physiology

P.S. - You can help your research get the attention it deserves! Check out Wiley's free Promotion Guide for best-practice recommendations for promoting your work at www.wileyauthors.com/eoo/guide. You can learn more about Wiley Editing Services which offers professional video, design, and writing services to create shareable video abstracts, infographics, conference posters, lay summaries, and research news stories for your research at www.wileyauthors.com/eoo/promotion.

• **IMPORTANT NOTICE ABOUT OPEN ACCESS:** To assist authors whose funding agencies mandate immediate public access to published research findings, The Journal of Physiology allows authors to pay an Open Access (OA) fee to have their papers made freely available immediately on publication.

You can check if your funder or institution has a Wiley Open Access Account here: <https://authorservices.wiley.com/author-resources/Journal-Authors/licensing-and-open-access/open-access/author-compliance-tool.html>.